Real-time biodiversity analysis using deep-learning algorithms on mobile robotic platforms

Panigrahi Siddhant sid.panigrahi1709@gmail.com
Maski Prajwal
Thondiyath Asokan
Department of Engineering Design, Indian Institute of Technology Madras , Chennai , Tamil Nadu , India
See Chan Hwang
Electronic publication date: 2023 Aug 25
Publication date: 2023
Volume: 9
Electronic Location ID: e1502
Received 2022 Jun 13; Accepted 2023 Jul 4
Copyright: ©2023 Panigrahi et al.
Copyright year: 2023
Copyright holder: Panigrahi et al.
License: This is an open access article distributed under the terms of the Creative Commons Attribution License, which permits unrestricted use, distribution, reproduction and adaptation in any medium and for any purpose provided that it is properly attributed. For attribution, the original author(s), title, publication source (PeerJ Computer Science) and either DOI or URL of the article must be cited.
License URL: https://creativecommons.org/licenses/by/4.0/

Keywords: UAV, YOLO v3, Deep learning, Object detection

Funding: The authors received no funding for this work.

==============================
Ecological biodiversity is declining at an unprecedented rate. To combat such irreversible changes in natural ecosystems, biodiversity conservation initiatives are being conducted globally. However, the lack of a feasible methodology to quantify biodiversity in real-time and investigate population dynamics in spatiotemporal scales prevents the use of ecological data in environmental planning. Traditionally, ecological studies rely on the census of an animal population by the “capture, mark and recapture” technique. In this technique, human field workers manually count, tag and observe tagged individuals, making it time-consuming, expensive, and cumbersome to patrol the entire area. Recent research has also demonstrated the potential for inexpensive and accessible sensors for ecological data monitoring. However, stationary sensors collect localised data which is highly specific on the placement of the setup. In this research, we propose the methodology for biodiversity monitoring utilising state-of-the-art deep learning (DL) methods operating in real-time on sample payloads of mobile robots. Such trained DL algorithms demonstrate a mean average precision (mAP) of 90.51% in an average inference time of 67.62 milliseconds within 6,000 training epochs. We claim that the use of such mobile platform setups inferring real-time ecological data can help us achieve our goal of quick and effective biodiversity surveys. An experimental test payload is fabricated, and online as well as offline field surveys are conducted, validating the proposed methodology for species identification that can be further extended to geo-localisation of flora and fauna in any ecosystem.

Introduction

Climate change is one of the most serious problems confronting the world, having direct or indirect impacts on all forms of life. Recent changes in environmental conditions, such as irregular rain cycles, flash floods, and global warming, in combination with anthropogenic acts of deforestation, pollution, poaching, and over-exploitation of natural resources, have all resulted in habitat loss and fragmentation of natural ecosystems. Encroachment on natural habitats has made the survival of indigenous species extremely challenging and has even led to the extinction of several endangered species. These irreversible changes in the ecosystem necessitate the need to track the biodiversity of a particular region, especially in dense wildlife areas which cater to most of the ecological biodiversity. These frameworks can help assess population density, monitor phenology variations, and research the overall species diversity of the mapped region. The data collected from such pipelines can also assist ecologists in making informed decisions about disease detection and diagnosis, species conservation, and habitat restoration for individuals in the modelled ecosystem. Identifying taxonomic species in any given ecology is one of the first steps towards such an ambitious goal of ecological modelling. Hence, the primary inspiration behind this study of biodiversity analysis is to take steps to reverse biodiversity loss by analysing data collected from individual ecosystems in real-time.

Conventional techniques of monitoring biodiversity for wildlife conservation are based on data collection by human field workers, which involve manual counting, observing, and monitoring individual animal species in a specific area. The presence of human field workers in such investigations not only endangers their personal lives but also has a negative impact on specimens present in the habitat. Biodiversity analysis and population forecast made from such capture, mark, and recapture techniques also lead to uncertain results because it is an extrapolation made from a small sample size to the entire ecosystem (Betke et al., 2008). Historically, we have a rich tradition of studying ecological data based on human understanding of species distribution and their diversity. Still, recent challenges in wildlife conservation have revealed such methods to be costly, time-consuming, and laborious (Witmer, 2005; Norouzzadeh et al., 2021).

Recent developments in low-cost and reliable sensing technology have paved the way for use of localised sensors in comprehensive ecological monitoring (Lahoz-Monfort & Magrath, 2021). These sensing techniques make use of both active and passive sensors for tracking diverse animal subjects in a specific area for a continual period of time. Such low-cost and precise sensory platforms have made monitoring possible in otherwise inaccessible areas and drastically reduced the cost of data collection. However, data acquisition strategies based on stationary sensor platforms are constrained when scaling up to large geographical areas. This is because species distribution is highly skewed in any ecosystem, and the deployment of sensing platforms must be increased to better understand the entire ecological area, adding to logistic and economic hurdles. Additionally, these sensor setups generate a reasonable amount of data on an individual basis. Still, collectively, the data piles up to be humongous, leading to a mismatch between the ever-growing dataset and the pipeline used to generate compelling ecological insights from such data.

Shortcomings in existing techniques lead us to two significant challenges in the biodiversity analysis task: data collection using portable, non-invasive, and low-cost remote sensing platforms, while the other bottleneck is providing a sound technological backbone to model and predict such complex, chaotic and ever-growing data. To address the data processing challenges, machine learning (ML) algorithms have been incorporated, which utilise statistical regression approaches to identify predictive models by detecting patterns in complex data. There have been instances where such models have been used in the past with supervised training sets for animal classification (Ferentinos, 2018), plant disease detection (Kutugata et al., 2021), and studying animal behaviour (Arablouei et al., 2023). However, conventional ML models do not rely only on labels in the dataset but specific properties of the animal subset (colour, pattern, and size), which need to be explicitly stated in terms of data (visual pixels or sound frequency). In contrast to the conventional ML models, deep learning algorithms (DL) can automatically extract and learn features from the dataset. Such an automated learning procedure is made possible by decomposing the data into multiple layers (artificial neural networks), which are connected to each other by different layers of abstraction (connection weights representing synaptic impulses) (Borowiec et al., 2022). Current approaches of automated biodiversity analysis using ML are also limited in their applications and acceptance because they do not consider three major biodiversity laws (Villon et al., 2022). Species, unlike inanimate objects, are difficult to classify, and any biodiversity monitoring approach should consider the three universal laws of biodiversity, namely: (a) skewness in distribution and abundance of the species, (b) species rarity across the ecosystem, which implies analysis of algorithms that can account for training on a scarce dataset, and (c) openness in the ecosystem which necessitates lightweight models that can be trained real-time on custom datasets considering the addition of ecologically similar species and elimination of critically endangered ones.

In this article, we demonstrate the application of light-weight DL algorithms for analysing custom-data collected from sample payloads mounted on autonomous robotic platforms to overcome the current challenges of data acquisition and dataset handling. The research objective here is to fabricate a mobile payload setup for data acquisition which can address the challenges of skewness in species distribution and overcome the limitations of stationary platforms. By choosing light-weight models from the existing state-of-the-art algorithms, which could run on such embedded payload setups, we could train the model on scarce datasets collected in wildlife settings and test the methodology for real-time applications. Furthermore, automating the entire data acquisition task with mobile aerial platforms can decrease the inherent cost by reducing the deployment of multiple setups while improving the quality of the collected data by adding spatio-temporal information. This computational framework can further be applied to any ecosystem, regardless of the region or size, by employing custom-made datasets that are tailored for species diversity in that area, accounting for the openness of the ecosystem. Interestingly, this methodology is still applicable for marine environments, which are primarily unexplored thus presenting opportunities for biodiversity mapping and conservation. Therefore, the primary contributions of this research focussing on an affordable and scalable biodiversity analysis framework are listed as follows:

1. Evaluation and comparative analysis of light-weight DL algorithms to detect and distinguish diverse animal species based on their performance characteristics in real-time wildlife settings.

2. Demonstration of the utility of an aerial platform and sample test payload for data acquisition and data-driven decision making for biodiversity analysis.

3. Experimental field testing using custom-built payload setup for onboard object detection in real-time wildlife settings.

The structure of the article is as follows: The “Background and Related Research” section compiles the existing methods used by researchers for wildlife tracking and lists their inherent drawbacks. The “Materials and Methods” section outlines the proposed methodology, which is then implemented on the collected dataset in the “Model Training and Validation” section. The final insights into the results and the scope of future work, along with its applications, are further discussed in the “Experimental Results” section.

Background and Related Research

The development of a cost-effective and scalable methodology for biodiversity monitoring is impeded by the two major challenges of data acquisition and dataset handling. Existing literature mentions several techniques utilised to collect ecological data for conservational and monitoring purposes. However, selecting the best data acquisition technique relies solely on the particular organism and mapped region being investigated. Similarly, numerous CNN (convolutional neural network), ANN (artificial neural networks), and RNN (recurrent neural network) architectures have been used in the past to generate inferences analysing the obtained data (Borowiec et al., 2022). As numerous different models yield different results based on the training dataset, which relies upon the data collection technique, and the architecture dictating the training duration and computing complexity of the entire process. This section briefly summarises the existing data collection techniques and the application of DL algorithms on such collated datasets to identify the intrinsic challenges in the current approaches, which can be addressed in this research.

Traditional methods for biodiversity analysis

The concept of using sensors for automated wildlife surveys is surprisingly not new, but rather an existing solution attempted by other researchers (Tuia et al., 2022). One of the primitive solutions to address wildlife conservation challenges is bio-acoustic sensing, which still finds its utility in many wildlife settings (Lostanlen et al., 2019). Animals use acoustic signatures for communication, mating, navigation, and territory marking. The well-established approach of bio-acoustic sensing makes use of microphones to record these sound sources, further enabling the classification of individual animal species based on their designated distinct acoustic signals. Since acoustic signatures present continuous time series data, important frequency parameters are extracted using Fast Fourier Transform (FFT) or RNN networks, which categorise data along with their associated temporal information (Chalmers et al., 2021). However, dense wildlife habitats are far more likely to be affected by background disturbances, making it necessary to have a pre-processing algorithm that can differentiate between the noise and the required ambient signal. The other drawback of bio-acoustic sensing is that it is only successful for some categories of animals, such as birds, bats, chirps, and insects (Kuenzi & Morrison, 1998). However, not all animal species; for example, predators produce regular acoustic signatures, making it difficult to detect and classify them.

The drawbacks of bioacoustic sensing shifted the emphasis from speech recognition to image classification, where trap cameras are used, triggered by the motion of animals to collect data. Visual data classification is easier to perform compared to auditory signals because of the richness in the collated data. However, the fundamental issue here is non-animal movements such as rain, thunder, and river flow, which also prompt garbage data collection. Besides these challenges, both systems, including bioacoustics classification or image processing using trap cameras, entail the deployment of sensors in poor weather conditions, demanding constant monitoring and maintenance of such platforms. The secondary issue is that stationary sensor setups capture localised data, which requires scaling such platforms for better coverage by increasing the deployment area across the entire ecological region. Stationary sensors also accumulate an enormous amount of data, demanding an efficient computational framework to process this colossal data. For instance, in the Snapshot Serengeti Project (Swanson et al., 2015), identifying the distribution of mammalian species in African Savannas necessitated the deployment of trap cameras in every 5 km2 area (leading to 225 cameras spanning across 1,125 km2 of Serengeti National Park) capturing close to 1.2 million images whose classification was made possible with the additional help of 68,000 volunteers. Some of the recent data acquisition initiatives using stationary setups along with their crowd-sourced datasets are highlighted in Table 1.

Despite having these conventional stationary platforms, the task of identifying individual species in an ecosystem is still very challenging due to the lack of generality across regions (species are not localised to a single place), time zones (day or night) and the need of manual labour (for annotation of datasets and maintenance of such setups). In recent times, some additional methods have also been employed, involving the use of invasive sensors for biodiversity analysis, as illustrated in Fig. 1. These methods include the use of miniaturized on-animal tags (Le Maho et al., 2011), incorporating RFID (Radio Frequency Identification), acoustic, and GPS (Global Positioning System) sensors used to monitor every single individual in an ecosystem. However, this approach of using on-animal tags is cost-intensive, considering that a separate setup is required for every individual in the ecosystem. Due to this limitation, these setups are mostly limited to livestock management and fish cultivation, where large-scale ecological predictions are not involved. Satellite imaging is also a relatively new technique helpful for biodiversity research, as it provides multi-spectral data for analysis that can be employed in various applications such as weather forecasting and estimation of vegetation cover. But satellite data is often collected at a constant GSD (ground sampling distance), reducing the collected data’s overall resolution. This sampled data is also susceptible to weather conditions, cloud cover and is updated at intermittent durations of time, making it difficult for continuous real-time monitoring. Therefore, aerial imagery from mobile robotic platforms often augments satellite remote sensing for extensive surveillance of vital areas at regular intervals and shows promising potential for automating biodiversity analysis.

Table 1 Recent wildlife conservation initiatives along with their sensing methodologies.

Sl. no.	Existing methods	Categorical classification	Sensing methodology	Overview	
1	Great grevy’s rally Parham et al. (2017)	Wildlife Census	Safari images	Census survey conducted by safari workers in Kenya to count the great zebra and giraffe species resulting in a repository of 50,000 images with 300 volunteers and 200 stationary sensor setups.	
2	BirdCLEF2019 Goëau et al. (2018)	Bird identification	Audio recorder	350 h of manually annotated time series acoustic recordings with 659 bird classes collected in Ithaca, US during 2017 and successively improved in 2019.	
3	Snapshot Serengeti Swanson et al. (2015)	Wildlife Sensing	Camera traps	Over 1.2 million images collected from 225 trap cameras in 1,125 square kilometres and 68,000 volunteers for classifying these images in Serengeti National Park.	
4	iNaturalist Computer Vision Van Horn et al. (2018)	Species identification	Crowd sourced species identification	Provides crowd-sourced datasets which can be used for classification models. Over 5 million images were collected and classified representing 73,000 species when the dataset was made public in 2017.	
5	Megadetector - AI for Earth Microsoft (2023)	Resource Conservation	Trap Cameras	Development of DL applications training on 900,000 trap camera images in 18 countries to facilitate ecological monitoring.	
6	Integrated Marine Observing System (IMOS) (IMOS)	Wildlife Sensing	Underwater sensory setups	Citizen science-based coral reef monitoring initiative with a network of 800 Argo floats, 27 autonomous underwater vehicles, 80 underwater moorings collecting essential ocean variables for conservation of Australian Great Barrier Reef.	

Figure 1 Existing methods for biodiversity analysis using sensor setups.

Deep learning for biodiversity conservation

Deep learning (DL) frameworks have perfused into multiple science disciplines because of their versatility and flexibility in handling intricate and enormous datasets. These neural network algorithms first gained attention in 2012 (Krizhevsky, Sutskever & Hinton, 2012), where it could generate meaningful inferences from audio signals and visual images by speech recognition and image classification. However, recent advancements leading to more robust and efficient neural networks have made these algorithms applicable to a broader range of real-world problems such as traffic surveillance (Tao et al., 2017), crowd monitoring (Zhao et al., 2019), biomedical image analysis (Kim et al., 2020), crop disease detection (Duhart et al., 2019), and rehabilitation tasks (Palani et al., 2022). DL algorithms have also been applied in wildlife settings to study pose detection and species identification. Focussing on plant disease detection, Mohanty, Hughes & Salathé (2016), demonstrated the classification of 14 crop-based diseases using a public dataset of 54,306 images comprising of healthy and diseased plants with an achievable accuracy of 99.35%. Another example of moth identification was provided by Bjerge et al. (2021), which identified eight different species of moths using an augmented dataset of 57,600 images and a maximum F1 score (referred in Eq. (2)) of 0.93. Few animal species identification projects have also been proposed with trap camera setups where Miao et al. (2019) discriminated 20 different mammalian species with an accuracy of 87.5% by training a VGG-16 and Resnet framework on a dataset consisting of 111,467 images. Similar frameworks have also been presented in distinguishing 20 coral species (Villon et al., 2018) with an accuracy of 78% in a dataset of 69,169 images and detecting vehicular traffic in adverse weather conditions (Hassaballah et al., 2020) using YOLOv3 architecture on public datasets like MS-COCO. As a result, this study aims to analyse and expand the use of such algorithms to include biodiversity analysis while collecting data from low-cost, non-invasive sensors mounted on a robotic platform.

Objectives of current research

The problem of scalability and cost-effectiveness, mentioned in existing techniques, can be addressed by the proposed method of developing a multisensory setup mounted on an unmanned aerial vehicle (UAV) capable of performing aerial manoeuvres for periodic and continuous data acquisition. Mobile robotic platforms, especially multirotors, show promising potential to provide a more reliable and compact platform for sensor integration in biodiversity analysis. By separating the mapped territory into a set of GPS coordinates and utilising waypoint trajectories to cover the targeted area, they can be utilised to collect data from any vast geography. The computational pipeline proposed in this study can then process this acquired data to create a wildlife activity map for real-time biodiversity analysis. Since multiple computational architectures have been used in literature, differing in the number of network layers and their individual composition. The aim is to evaluate the optimum framework for such a real-time application by training different state-of-the-art algorithms and bench-marking their performance metrics.

Materials and Methods

In comparison to the existing methodologies, the use of mobile platforms can address the problems of skewed species distribution by its mobility, as well as overcome the limitations of increased deployment of stationary sensors for better ground coverage. However, several critical challenges are also introduced, such as battery limitations, sensor shortcomings, sampling rates of sensor platforms, geographical footprints of these aerial robots, onboard memory requirements, and selection of computationally efficient algorithms which must be thoroughly evaluated for such real-time applications (McEvoy, Hall & McDonald, 2016; Panigrahi, Krishna & Thondiyath, 2021). This section provides an overview of the proposed methodology adopted to develop the real-time biodiversity map using aerial imagery to address some of the above-mentioned challenges. The complete task can be divided into four primary steps: (i) fabrication of embedded setup and mobile platform for data collection, (ii) dataset construction, image pre-processing, and labelling of collected data by annotation, (iii) object classification to detect the animal subset in real-time, and (iv) performance evaluation in real-time wildlife settings. The remainder of this section discusses the aforementioned tasks sequentially in order of their implementation.

Hardware implementation for data collection

A custom UAV is fabricated with the aluminium skeletal framework in Quadcopter-X configuration outfitted with 400 KV brushless DC motors and 12” × 8” carbon fibre propellers for propulsion. Visual data sampling is performed by an embedded payload setup mounted on the pan–tilt gimbal mount of the mobile robotic platform as illustrated in Fig. 2. Autonomous waypoint trajectory control is implemented on Pixhawk 4 Flight Controller to manoeuvre across the planned GPS coordinates spanning the region of interest. Simultaneously flight telemetry is performed by Raspberry Pi4 (single board computer) with a Zigbee Transmission module to transmit flight data parameters and sample payload data to the ground control station.

This airborne platform is powered by a 14.8 V, 5200 mAH lithium-polymer (LiPo) battery, enabling the flight endurance of 20 –25 min based on wind gust disturbances and the gross weight of sample payloads. The overall flight payload is developed with the intent of performing both online (through the onboard computation) and offline computations (via the ground control station) for biodiversity analysis. These sample payloads also varied for both the modalities, as the online computation required an onboard computational setup and vision sensor (Nvidia Jetson Nano 4GB with 128 CUDA Cores for onboard analysis along with 8 Megapixel Sony IMX219 CMOS sensor of Raspberry Pi V2 camera), while the offline computation relied on the vision sensors (23.6 Megapixel GoPro CMOS sensor with a field of view of 122.6°) along with the flight telemetry (Zigbee Xbee 2mW transmission antenna) transmitting to the ground control station (16 GB Intel i9-10900 processor with 8 GB NVIDIA GeForce RTX 2060 Graphics Processor). The entire framework highlighting both the hardware schematic and computational pipeline is depicted in Fig. 2.

Figure 2 Schematic of the proposed methodology.

Computational pipeline framework for biodiversity analysis

The two key requirements for the computational pipeline in biodiversity analysis are object detection and target tracking of the detected object. Currently, algorithms for object detection are broadly classified into two categories: a two-stage detection model, which selects candidate regions of interest from an image and then classifies these regions like Faster-RCNN, then there are one-stage detection models like YOLO, SSD, and DeepSort which use regression techniques to minimise the classification and localisation error functions. In comparison of these two models, one-stage detection models are faster and more efficient than two-stage detection models because they simultaneously execute both object detection and tracking tasks.

Single-stage detection models are proved to be more efficient, since the neural network predicts the bounding boxes of the designated class and its associated confidence in a single evaluation. Since all the predictions are made in a single flow, multiple improvements have been made to increase these algorithms’ detection speed and accuracy, which involve optimisation in the architecture of the constituent layers. But in order to achieve efficient real-time object detection, these modified algorithms require the usage of high-performance GPUs. When these efficient algorithms are tested in embedded platforms employed in mobile robots, the object detection speed substantially reduces. The primary reason for this decrease is the performance of the GPU on the embedded platform, which is comparatively lower than that of a workstation contributing to the increase in the processing time.

Secondly, aside from the computational framework performance, the hardware setup required to accelerate state-of-the-art DL models is also dictated by the computational network architecture and design requirements of the mobile platform. While selecting the DL algorithm for a particular task, consideration of the algorithm’s memory and power requirements is crucial. Recent DL models like SSD, YOLO, and Deep Sort have memory requirements (in Flops, floating point operations needed to be computed per second) much larger than the standard embedded device memory sizes. Thus, to address the memory constraints, a lesser complex backbone network for feature extraction should be considered, which has fewer parameters such that they can fit within the available memory of the device. To further address the bottleneck of energy-efficiency, which is primarily due to the intensive computation carried out, multiple hardware platforms should also be compared for their intrinsic performance and suitability. As a result, the entire problem reduces to selecting algorithms that can handle model structure complexity and the selection of a hardware platform to address power consumption and overall weight.

The first problem of model complexity in the architecture can be addressed by comparing the performance of lightweight convolutional models, such as Yolo-Tiny (Maski & Thondiyath, 2021), Mobile-Net (Howard et al., 2017), and Squeeze-Net (Ucar & Korkmaz, 2020) that are designed for applications that fit the needs of mobile and embedded devices with other state-of-the-art algorithms, as mentioned in Table 2. This list includes different single-stage and two-stage detection models from the past decade, varying in their backbone architecture, computational memory requirements, and differing in their embedded size. It can be easily noticed that with an improvement in performance (in FPS) both the memory requirements (in Flops) and required embedded size (in MB) also increases. Hence extensive analysis of the performance metrics of these algorithms should be studied for species identification in real-time wildlife settings, which are rarely investigated. Thus in this subsequent section, we present a comparative analysis of light-weight convolutional models along with the standard DL architectures listed in Table 2, to investigate the trade-off of performance and accuracy in such real-time applications.

Table 2 Comparison of state-of-the-art DL algorithms analysed in this study.

Deep learning framework	Backbone architecture	Memory requirements	Ideal speed	Embedded size	
Faster RCNN (2015)	VGG-16	7 BFlops	20 FPS	134.5 MB	
SSD- MobileNet-v2 (2017)	VGG-16	4.3 MFlops	32 FPS	4.5 MB	
YOLOv3 (2018)	Darknet-53	65.86 BFlops	62 FPS	236 MB	
Tiny-YOLOv3 (2018)	Darknet-19	5.56 BFlops	220 FPS	33.4 MB	
MobilenetV2- YOLOv3 (2017)	MobileNetV2	2.44 BFlops	244 FPS	11.52 MB	
YOLO-Fastest (2021)	NCNN	0.25 BFlops	230 FPS	1.3 MB	
YOLOv5x (2021)	CSPDarknet-53 (Cross stage Partial)	222.9 BFlops	83 FPS	166 MB	
YOLOv7 (2022)	YOLOv4, Scaled YOLOv4, and YOLO-R	104.7 GFlops	161 FPS	36.9 MB	

Model training and validation

This section summarises model training and explains the different benchmarks used to quantify the performance of proposed DL frameworks. The initial step is to generate the custom dataset, train different neural network models on a virtual colaboratory platform, and extract multiple indicators to benchmark their applicability in wildlife surveying.

Dataset creation and annotation

A self-collated public dataset of 881 images of different deer species is used in this research. This dataset comprises multiple images collected from aerial imagery, trap cameras, sample payload setup fabricated in this study, and handheld cameras. In any biological dataset, there is a persistent problem of long-tailed distribution which implies that there is always a sparse distribution of animal target objects in the collected sample images.

This can be further verified from Fig. 3, which illustrates a lack of data with multiple target objects in it (more than 11-16 deers) and the annotation heatmap which also shows concentrated information towards the centre of the annotated visual data. Hence data augmentation is performed by applying spatial transformation on the sample data to increase the dataset to 2,100 images, and image duplicates are also added. The median width and height of the sample images collected are 1024 px × 720 px, indicating that all images are of irregular scales and sizes, requiring scaling to 416 px × 416 px prior to model training. The distribution of the annotation across the images is illustrated via labels correlogram in Fig. 3D. It can be seen from the subplot of (X, Y) that the labelled parts cover the entire image, and the parameters are normally distributed. But from the subplot of (Width, X), (Width, Y), (Height, X), and (Height, Y) it can be observed that the labels are unevenly distributed with most of them under the ratio of 0.4, indicating that dimensions of most of the annotations are small enough and do not span across the entire width and height of the image. To address this unevenly distributed dataset, augmentation is performed and other images from Google open image dataset v6 are added (Google, 2021). Subsequently, manual annotation is performed on the sample images where the bounding boxes are provided, and classes are assigned using LabelImg (HumanSignal, 2021) as shown in Fig. 4.

Figure 3 Aspect ratio comparison, annotation heatmap, and sample object distribution in the custom dataset.

Figure 4 Manual annotation of sample images into classified classes by drawing the bounding box.

Model training

The proposed DL neural networks are trained on a virtual machine platform utilising Google colaboratory server, which enables enhanced operational runtime with NVIDIA Tesla K80 GPU. Standard DNN networks and proposed lightweight frameworks are trained on the custom dataset with 80%–20% split for 6,000 epochs at a learning rate of 0.001 and batch size of 64 to account for the memory requirements of the server. The initialisation parameters of the proposed model, such as momentum, initial learning rate, and weight decays, are extracted from the original YOLOv3 model as tabulated in Table 3.

Table 3 Initialisation parameters for the proposed learning model.

Size of input image	Batch size	Training iterations	Momentum	Initial learning rate	Decay	
416×416	64	6,000	0.9	0.001	0.0005	

Evaluation parameters for the framework

To quantify the performance of multiple algorithms proposed in this study, various performance metrics are derived based on the training of the algorithms on the training dataset and results from the validation dataset. Two major indicators are extracted to measure the rate of learning and prediction accuracy of such frameworks to detect the target animal subset, namely:

(a) Training loss (L): The first evaluation metric is the training loss which indicates the rate of learning of the model as the training cycle progresses. Overall loss function in any object classification problem is generally expressed as the algebraic sum of three individual categories of losses as depicted in Eq. (1): (1) L=λconfLconf+λclaLcla+λlocLloc

where λ is the corresponding weight assigned to the individual losses, Lconf is the target confidence loss which measures the objectness or correctness in the individual prediction, Lloc is the coordinate prediction error arising due to mismatch in the anticipated bounding box and ground truth label, and Lcla is the target classification loss that measures the error in the prediction of classified category. Training loss in any neural network quantifies the difference between the ground truth label’s expected output and the neural network’s predicted output. Faster convergence in the training loss during model training not only implies a better learning rate of the algorithm but also guarantees training in lesser epochs dictating the scalability of the model.

(b) Mean average precision (mAP): The second evaluation metric is mAP which indicates what fractions of the predictions are accurate. Loss metrics are helpful for analysing the performance of the models during training and for determining the amount of time and dataset diversity needed to train the model effectively. However, classification metrics are utilised to analyse the correctness of predictions made by such a framework.

To extract the classification metrics, a confusion matrix is constructed, which is a tabular visualisation of the neural network’s predictions illustrating the performance based on four prediction categories. The four categories are TP (True Positive) represents the number of objects (in our case deers) that are actually present and have been correctly identified, FP (False Positive) implies that the object is actually not present but still falsely identified, FN (False Negative) represents the number of objects that are actually present but still not identified, and TN (True Negative) which implies that the object is actually not present and the model classifies neither. We represent the confusion matrix based on these categories of predictions made on the validation dataset, such as TP, FP, TN, and FN. Classification metrics like Precision, Recall and F1score are further derived from the confusion matrix as expressed in Eq. (2): (2) Precision=TPTP+FP,Recall=TPTP+FN,F1score=2Precision×RecallPrecision+Recall.

Here Precision is the fraction of correct predictions out of all actual positive cases. Recall is defined as the ratio of actual positive cases which are correctly identified out of all total correctly identified predictions. On the contrary, F1score combines both the Precision and Recall values into a single metric for the classifier model by taking the harmonic mean and giving equal weightage to both of these values to evaluate the classification performance.

Following the evaluation of precision and recall values for individual algorithms in each iteration, the Precision-Recall curve is plotted with precision on the vertical axis and recall on the horizontal scale. The area under the P-R curve is calculated to obtain the average precision (AP). This average precision is further calculated for each class and later, the mean of all such classes is calculated to obtain the mean average precision (mAP) as expressed in Eq. (3). (3) mAP=1Nclasses ∑n∈classesAPc.

Apart from precision and recall, which are primarily derived from the confusion matrix, there is another evaluation standard called IoU (Intersection over Union). IoU is the metric to obtain the detection accuracy by calculating the ratio of overlap between the predicted bounding box and the ground truth label as expressed in Eq. (4). (4) IoU=SoverlapSunion,

where Soverlap is the intersection between the predicted bounding box and the ground truth label, and Sunion is the union of the two bounding boxes. These extracted classification metrics like F1score, IoU, and mean average precision (mAP) provide us with a measure of how correct our predictions are irrespective of the category or class, which dictates the generalisability of the model while the training loss metrics define the scalability of the model.

Comparative analysis of the algorithms

Performance metrics, defined in the previous section, such as training loss, F1score, mAP, detection speed, and sample image inference time are compared to quantitatively rank the algorithms for applications in real-time wildlife surveying. The variations in the training loss as the learning iteration progresses are plotted in Fig. 5. Here it can be easily observed that two-stage detection models (Faster RCNN) perform poorly in comparison to other single-stage detection models (Mobilenet SSD-v2, YOLO), having the least gradient in training loss among all other standard frameworks.

Figure 5 Parametric comparison of trained algorithms based on the training loss and mean average precision.

Among other single-stage detection models, as illustrated in Fig. 5A, the training loss converges after 500 iterations in YOLO v7 and 2500 iterations in YOLO v5 where the optimiser stripped the training cycle since there is no significant improvement in the overall loss function after 2,000 iterations. In contrast, the training loss converges for 3,000 iterations for the lighter version of YOLOv3, while in the standard YOLOv3 model and Mobilenet SSD-v2 it saturates after 6,000 iterations. Even on comparison of the magnitude of training loss after 6,000 training cycles, the training loss of Mobilenet SSD-v2 (1.002) and YOLO v3 (0.612) was substantially larger than recent versions of YOLO v7 (0.177), followed by YOLO v5 (0.027) and proposed lighter version of YOLOv3 (0.10 or less than 0.10).

Training time is another important parameter considered for algorithm comparison as it infers on the model complexity. In the comparison of training time, YOLO v7 requires 43.5 hours, followed by YOLO v5 (30.912 hours), YOLO v3 (10 hours), while Mobilenet SSD-v2 and other lighter version of YOLO v3 requires less than 3 hours to train for the same number of iterations. The major conclusion is that the training time is directly proportional to memory requirements which dictate the number of parameters to be computed and the complexity of the neural network. Here it can be concluded that recent versions of YOLO (namely YOLO v5 and YOLO v7) perform comparatively better than YOLO v3, but at the cost of model complexity leading to greater training time and increased memory requirements, as tabulated in Table 2. However, lighter frameworks of YOLO v3, owing to their less model complexity and better convergence in training loss within fewer training iterations, demonstrate a better learning rate and faster inference time during training with the sample dataset.

Another notable observation from Fig. 5, is the inverse relationship between the training loss and mAP. As the training loss converges, the mAP increases due to a reduction in localisation, classification, and target confidence loss, which is self-explanatory. From Fig. 5B, it can be noticed that both Faster RCNN and Mobilenet SSD v2, which have the least learning rate settle at a lower mAP (≈75%) in comparison with the other algorithms. Recent versions of YOLO have a faster learning rate but settle at a comparatively lower mAP (≈94%) due to no significant improvement in the learning rate after 2500 iterations. Among other DL frameworks, both YOLO v3 (99.25%) and light-weight model Tiny-YOLO v3 (99.35%) achieve the best mAP for the same training iterations. It can also be depicted that most of the two-stage detection models excluding Mobilenet SSD-v2 demonstrate a similar trend in mAP, which is almost 95% for a confidence threshold of 0.25. In simple words, it implies that all the algorithms are equally capable of anticipating the ground truth label with considerable accuracy. This can be further verified from Fig. 6, where in-depth visualisations of the predictions are presented. It can be concluded that most of the proposed algorithms are able to detect the animal subset with comparable accuracy, whereas lightweight frameworks require the least training time. As a result, memory requirements, detection speed, and training time are considered as the primary evaluation metrics for determining the most suitable algorithm when they all are equally accurate owing to similar mAP.

Table 4 presents all the extracted performance indicators, with the best parameters highlighted in green, the second-best in yellow, and the least effective in red. From the statistical comparison of all the available performance metrics as shown in Table 4, it can be concluded that the lightweight models, more specifically Tiny-YOLO v3 clearly outperformed the conventional DL frameworks in terms of learning rate (81.72% reduction in overall average training loss), detection accuracy (9.76% improvement in average mAP), and sample inference time (76.63% improvement in comparison to the average detection time with YOLO v5 having similar inference time but greater training time and model complexity). Out of all the proposed algorithms, Tiny-YOLOv3 has a better learning rate with a training loss of 0.1155 for 6,000 training iterations and was able to detect the animal subset in about 15.8 ms at a detection speed of about 32 FPS with support of NVIDIA Tesla K80 GPU, proving to be suitable for wildlife surveying applications.

Figure 6 Confusion matrix for the proposed algorithms.

Hence lighter versions of YOLOv3, owing to their simpler backbone structure, lower memory requirements, and greater learning rate, are found to be more computationally efficient for real-time biodiversity tasks. However, their practical feasibility in wildlife surveying in real-time scenarios must be studied, which is discussed in the upcoming sections.

Experimental Results

This section presents the result from experimental testing including indoor lab evaluation (from the results of validation dataset applied to in-situ environments) and real-time testing (utilising field surveys conducted by the fabricated setup). The custom animal dataset is constructed using sample visuals captured in wildlife settings which is further split-head in 80%–20% ratio, where 80% of the collected data is used for model training, and the remaining 20% is used as a test dataset to evaluate the performance of the trained algorithms.

Table 4 Performance metric chart of the proposed DL algorithms.

Training model	Mean average precision (mAP)	F1 score	IoU	Minimum loss after 6,000 iterations	Detection speed (FPS)	Time spent in detection (ms)	
Faster RCNN	69.90%#	0.70#	–	2.995#	17	88.43	
SSD-MobileNet-v2	76.95%	0.78	–	1.002	10#	132.4#	
YOLOv3	99.25%$	0.99@	92.03%@	0.6121	13	104.6	
Tiny- YOLOv3	99.35%@	0.99@	87.03%	0.1155	32@	15.8$	
MobileNetV2 - YOLOv3	97.69%	0.97$	81.15%	0.023@	18	69.04	
YOLO-Fastest	96.31%	0.93	74.82%#	0.1016	16	80.43	
YOLOv5x	90.44%	0.88	81.1%	0.0275$	30$	14.8@	
YOLOv7	94.24%	0.89	87.3%$	0.1776	28	35.71	
Notes.

The best parameters are denoted by “@”; the second-best parameters are denoted by “$”, and the least effective are denoted by “#”’ symbols.

Figure 7 Results of the proposed trained algorithm identifying the animal subgroup (Deer) based on the validation dataset.

Indoor laboratory testing

Multiple algorithms trained on the Google colaboratory platform are evaluated on the validation dataset to ensure their robustness in various natural scenarios. The training dataset encompasses visuals exploring varying environmental conditions (low ambient light, occlusions, and camouflage) and includes numerous specimens of the same species and individuals from other species for better training and performance in field surveys. As shown in Fig. 7, the results of the validation dataset provide further evidence on the application of trained algorithms in a wide range of environmental scenarios. The next step was the evaluation of the most efficient algorithm for such a biodiversity analysis task. Multiple performance metrics are compared to identify the algorithm’s efficiency in detecting the target dataset with the highest accuracy in the least amount of time and with minimum computational resources. According to the comparison of all the available algorithms given in Table 4, Tiny-YOLOv3 emerges as the most effective algorithm for detecting the animal subset in the validation dataset.

Following the identification of the most efficient computational framework, two additional challenges must also be addressed. The primary challenge is determining the best hardware components to accelerate the algorithm’s performance while still being light enough to be used as a sample payload for airborne platforms. While the secondary challenge is testing the performance of the same algorithm in real-time applications on an embedded setup. The computational framework was selected based on the multiple performance metrics extracted from the results of the validation dataset. But the platform selection is majorly dependent on power consumption, inherent cost, overall weight, and maximum frames it can analyse for real-time applications. An inherent comparison of the existing hardware platforms is provided in Table 5. The derived parameters are verified by the fabricated indoor lab setup, as shown in Fig. 8, while other functional specifications are extracted from the existing literature (Mazzia et al., 2020; Meng et al., 2020). The proposed normalised feasibility score (Fn), as depicted in Table 5, is computed to identify the best platform suited for such studies. Higher normalised feasibility score (Fn closer to 1) indicates that the selected platform is lighter, cheap, and consumes less power while achieving greater FPS when used in real-time DL applications.

Table 5 Comparison of the different hardware platforms for their utility in real-time applications on mobile robotic platforms.

Hardware platform	Price (A) in US dollars	Maximum achievable FPS (B)	Maximum power consumption (C) in watts	Overall weight (D) in grams	Normalised feasibility score Fn =[B/(A+C+D)]/Fnmax	
Intel Movidius + Raspberry Pi3	70	4.8	5.6	95.36	0.765	
Intel NCS 2 + Raspberry Pi3	74	5	6	107.13	0.728	
Jetson Nano	99	8	19	100	1@	
Jetson AGX Xavier	800	30	30	274	0.7404	
Notes.

Fn = 1, denoted by @ symbol.

Figure 8 Experimental test setup for indoor testing. Here subfigures (A) illustrate the exploded view of the first version of the test setup and (B) depict the real-time output of the detection model.

Figure 9 Results of the Tiny-YOLOv3 algorithm on sample aerial footages.

Bench-marking of the hardware specifications presented in Table 5, reveals that commercially available neural sticks and hardware USB accelerators (Intel Movidius, Intel NCS) are lighter to mount on an aerial platform with less power consumption but have very nominal FPS restricting their usage in real-time applications. On the other hand, Jetson AGX Xavier, with Volta GPU micro-architecture and 64 Tensor Cores, can reach computational speeds upto 11 TFLOPs demonstrating superior performance, but at the cost of power consumption and increased payload weight, sacrificing the endurance of the mobile robot. As a result, the Jetson Nano platform is chosen, which has the highest normalised feasibility score (Fn = 1) among the available onboard embedded devices. It is powered by an NVIDIA Maxwell GPU with a maximum computational capability of 472 GFlops, has a nominal weight of 100 grams, consumes approximately 5–10 watts of power in real-time operations, and can provide an ideal achievable detection speed of 8 FPS, making it the ideal choice for such lightweight setups.

The fabricated sample test setup implemented for indoor biodiversity analysis is demonstrated in Fig. 8. Multiple versions of the test setup are also fabricated to address the constraints of weight and power consumption, making it a sample payload for mobile platforms. In addition to testing the algorithms on stand-still images, attempts are made to compare their performance in aerial imagery and real-time footage obtained from the setup. Results of implementation of the Tiny-YOLOv3 algorithm on sample aerial imagery collected from the commercially available DJI Mavic platform (https://www.youtube.com/watch?v=92MfFcyjRSQ) is demonstrated in Fig. 9 at different intervals of time.

However, during the evaluation of the indoor setup in offline mode, most of the data processing is performed in the ground control station or via offline methods. The primary difference between offline and online implementation is the results from the offline detection are not real-time. Hence the species abundance and distribution are from the instant of time when the aerial imagery was performed. Hence online implementation utilising resource-limited setups in aerial platforms should also be explored, providing real-time results as data processing and collection are simultaneously performed using the onboard processors. These preliminary findings demonstrate the potential of aerial imagery as a practical resource for biodiversity conservation. However, there is still a need for a more in-depth analysis of the benefits and drawbacks of both online and offline deployment, which is discussed in detail in the following section.

Real-time biodiversity analysis

In this study, field investigation for real-time wildlife identification is carried out in the Crocodile and Snake Bank, a captive breeding site located in Guindy National Park (Tamil Nadu Tourism, 2021). This wildlife conservation park is home to diverse species of plants and animals due to its situation on the eastern subtropical banks of Chennai, as visualised geographically in Fig. 10. The target animal subset for in-situ laboratory testing was deer, which included multiple species of Spotted deer, Blackbuck, and Sambar deer. However, to validate the feasibility of the setup in wildlife settings, we consulted with the wildlife ecologists from Guindy National Park to choose an alternative target animal subset.

According to the point of view of field experts, the entire spectrum of species that are evaluated for the biodiversity census can be categorised into four major classes: birds, herbivorous mammals, carnivorous mammals, and lastly reptiles. While birds are abundantly found, herbivorous mammals are easy to tag and locate. The major challenge arises in detecting and tracking carnivorous mammals and reptiles that excel in camouflaging within the environment. Therefore, for the final field testing, the selected target animal subset is Crocodile belonging to the class Reptiles. The immobile nature of this species makes it hard to detect, making it a topic of interest for such wildlife detection tasks. Although multiple species of crocodiles are found in the proposed region of study, such as Gharials, Freshwater crocodiles, Mugger crocodiles, and Snouted crocodiles, we focused on the complete animal subset rather than the individual species.

Here we focused on the distribution of the target subset regardless of the species leading to the creation of a several dataset comprising 324 images of several Crocodile species. The selected DL-based Tiny-YOLOv3 algorithm is trained on a custom-made dataset consisting of 324 sample images of the target animal subset, including all the major crocodile species present in the study region. Data acquisition is conducted from the test setup at different intervals of time. Similar images are also added from Google’s open image dataset to add to the richness of the collated dataset. The modified test setup is improved from the existing setup implemented in the indoor laboratory settings (Panigrahi, Maski & Thondiyath, 2022). Minor modifications are incorporated into the current design to reduce the overall weight (446.2 grams), making it to be lightweight and compact enough (10 × 14 × 3.5 cm) to be mounted on the airborne platform. The camera setup was also upgraded to a Raspberry Pi V2 camera, and the outer protective acrylic case was fabricated to make it into a mountable sample payload setup, as shown in Fig. 11.

Figure 10 Geographical visualisation of the area under study is Guindy National Park, situated in Chennai, Tamil Nadu.

The geographical maps are created using open source software QGround control (Dronecode, 2021) which is used for geo-localisation of the Pixhawk flight controller.

Figure 11 Test setup employed for real-time biodiversity analysis.

Here, subfigure (A) illustrates the modified setup mounted on the tripod stand, (B) depicts the overall weight of the test payload, (C) shows the enlarged view of the modified setup, (D) shows the mobile aerial platform with the sample payload, and (E) depicts the validation using the ground control station.

Figure 12 Test setup deployed in the crocodile bank of Guindy National Park.

The fundamental objective of this research is to demonstrate the feasibility of the proposed hardware and computational platform for application in wildlife settings. The embedded platform with onboard sensors is built, and real-time species detection using sample payload setup is performed on the stationary mount as shown in Fig. 12. Nevertheless, the setup can also be mounted and implemented on the mobile robotic platform to perform automated field surveys as validated earlier in Fig. 9. The successful real-time implementation of the test setup in wildlife settings is illustrated in Fig. 13. Experimental results verify that the algorithm can recognise the target subset with reasonable accuracy, and the methodology can also be applied to a wide variety of plant and animal species. The major conclusion from both the online and offline implementation is the reduction in the processed frames, which drastically reduced from 32 FPS in offline implementation to 4 FPS in online implementation due to onboard processing. However, this latency can be considered as a tradeoff since the entire data is streamed in real-time, enabling the ecologists to make data-driven decisions on the go.

Figure 13 Results from the real-time output of the test setup in the proposed area of study.

Conclusions

This research investigated the use of state-of-the-art DL algorithms in wildlife surveying for real-time biodiversity analysis. The algorithms were trained using visual data of multiple species in various wildlife settings using off-the-shelf cameras and aerial imagery. The diverse dataset comprises of images that were collected and manually labelled for training the algorithms and testing its generalization for multiple species in different environments. In practical scenarios, multiple DNN (deep neural networks) algorithms were evaluated based on their performance metrics in object detection and object tracking tasks. It was finally concluded that the lighter versions of YOLO, specifically Tiny-YOLOv3 outperformed the standard neural networks in terms of average precision (mAP of 99.35%) and detection speed (32 FPS for online and four FPS for offline). An embedded device setup was built to demonstrate the real-time implementation of these algorithms on mobile robotic platforms. Multiple hardware platforms were compared, and the final arrangement of the embedded device setup was laid down, comprising of Jetson Nano onboard computer and Raspberry Pi-V2 camera for image acquisition. The fabricated embedded setup resulted in a compact payload arrangement that is small (10 × 14 × 3.5 cm), lightweight (446.2 grams), consumes less power (5–10 Watts), and is capable of executing DNN algorithms on mobile platforms in real-time applications.

Initial research and experimental validation demonstrate the efficacy of DL algorithms in wildlife surveying and mobile robotic platforms for ubiquitous data collection. Nevertheless, our future goal is to incorporate spatiotemporal information such as location, time, habitat, and weather alongside the visual object detection outputs to better understand the population dynamics of an ecosystem. The primary goal of this study is to determine if low-cost vision sensors combined with DL algorithms could facilitate automated wildlife biodiversity mapping. Further advancement of this research is to examine how different types of sensors (thermal, GPS, and IR vision) can be used together to acquire data and how this information affects the specificity of real-time object detection. Object detection tasks can also be improved to make concurrent behavioural detection of the predicted animal class (i.e., standing, resting, pregnant, feeding, grooming) with the help of pose estimators. The current version of the proposed pipeline can detect multiple animal datasets in practical wildlife settings. However, with the above-mentioned enhancements, this framework offers a scalable, cost-effective, and efficient solution to collect and analyse unobtrusive data in real-time environments. This data can prove to be crucial for ecological restoration by improving our understanding of species distribution and our ability to predict how these specimens will respond to the constantly changing environmental and anthropogenic factors.

Supplemental Information

Supplemental Information 1 Results on the collated dataset.

Click here for additional data file.

The authors are thankful to Mr. R Rajarathinam and the entire Gunidy National Park administration for their continual assistance, domain expertise, and advice for this project. The first author would also like to thank Professor Asokan Thondiyath for the conceptualisation of the idea, Sumitra Panigrahi, Rabi Narayan Panigrahi, Swayamsiddha, and Prashant Kumar for their support in bringing this idea to life, and members of Robotics Lab, IIT Madras for their immense help and support during the course of this project.

Additional Information and Declarations

Competing Interests

Author Contributions

Data Deposition

The authors declare that there are no competing interests.

Siddhant Panigrahi conceived and designed the experiments, performed the experiments, analyzed the data, performed the computation work, prepared figures and/or tables, authored or reviewed drafts of the article, and approved the final draft.

Prajwal Maski conceived and designed the experiments, performed the experiments, analyzed the data, performed the computation work, prepared figures and/or tables, and approved the final draft.

Asokan Thondiyath conceived and designed the experiments, performed the computation work, authored or reviewed drafts of the article, and approved the final draft.

The following information was supplied regarding data availability:

The code file, images, and a visual representation for the implementation of the project

The data and code are available at Zenodo: Siddhant Panigrahi, Prajwal Maski, & Asokan Thondiyath. (2023). Real-time biodiversity analysis using deep-learning algorithms on mobile robotic platforms. https://doi.org/10.5281/zenodo.7934912.

The video is available at YouTube:

https://www.youtube.com/watch?v=5o6jpqLJW5Y.

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
