# Peer review of "Real-time biodiversity analysis using deep-learning algorithms on mobile robotic platforms"

_PeerJ Computer Science, doi:10.7717/peerj-cs.1502_

## Round 0.1 · original submission · Major Revisions

Based on the comments of both reviewers, I recommended "Major revisions" for this paper. Authors should carefully address all the comments from the reviewer in particular to enhance the discussion of experimental results and setup.

Reviewer 1 ·

Basic reporting

The paper proposes a deep learning-based biodiversity analysis method using the aerial platform (UAV) instead of stationary platforms in real-time. They created a testbed to train and test the proposed method, onboard object detection, and evaluate its performance

Experimental design

Biodiversity analysis is important for mitigating the impact of climate change on the ecology and taking steps to reverse biodiversity loss by analyzing data collected from individual ecosystems in real-time. Traditional biodiversity monitoring techniques have challenges, such as utilizing stationary platforms for data acquisition and continuous updating. Moreover, the current studies rely on the census of the individual population using the capture, tag-mark, and recapture technique, which is labor-intensive, expensive, and time-consuming. This study proposed the deep learning-based method implemented on high-resolution spatial images or live footage using the aerial platform (UAV) for creating a real-time biodiversity map to address the above challenges. A testbed is designed to train and test the proposed method and onboard object detection. The authors provided outdoor and indoor test results to evaluate the proposed solution for real-time biodiversity analysis.

Validity of the findings

The idea of the paper and the proposed solution is convincing. However, this paper still has some flaws; hence revisions are required. The suggestions are listed as follows: - The authors stated that data acquisition and handling are challenging in stationary platforms, and they proposed a solution using the aerial platform to overcome their current challenges. However, the challenges of data analysis in stationary platforms are not clear. The challenges need to explain in detail. - It is mentioned that the quality of the collected data will improve by automating the entire data collection task with aerial platforms. This part is not clear. If the said the collection of high-quality images, it could also be collected through fixed platforms. It can be explained in more detail. - The state-of-the-art image recognition works did not address in the existing research section. This section can be extended by adding state-of-the-art image recognition works. - In this work, the YOLO algorithm is used for biodiversity analysis. There are many deep learning algorithms for real-time object detection such as R-CNN, R-FCN, etc. It is not clear why the YOLO algorithm was chosen. It can be clarified in more detail. - The specification of the dataset can be added. Is the training done on the same dataset for outdoor and indoor testing, or is a different dataset used? It also can be added to prevent confusion. - The complexity analysis of the proposed solution can be added. - Authors are suggested to improve the grammar and sentence structure across the article.

·

Basic reporting

no comment

Experimental design

no comment

Validity of the findings

no comments

Additional comments

the manuscript cannot be considered for publishing at PeerJ Computer Science for several reasons such as:

the abstract need to be rewritten.

the literature review of the state of the art is very weak and did not include recent trends in Object detection.

the experimental results are not enough, special the authors mentioned in the introduction section that they will prepare an evaluation and comparative analysis of deep learning algorithms for detection part but unfortunately the proposed method is not compared with recent published methods in range 2020-2022 while only different configurations of YOLOv3 are included in EXPERIMENTAL TEST RESULTS section.

---

## Round 0.2 · accepted · Accept

Authors have addressed all the comments from the reviewers. The paper is recommended to be accepted in its current form.

Reviewer 1 ·

Basic reporting

the paper is in acceptable format.

Experimental design

the paper is in acceptable format.

Validity of the findings

the paper is in acceptable format.

Additional comments

the paper is in acceptable format.

·

Basic reporting

See below

Experimental design

See below

Validity of the findings

See below

Additional comments

I have finished the review. I found all my comments have been answered by the authors and the last version has been updated taken into account my recommendation. The submission can be accepted to publish.